# Structural and Functional Analysis of ASFV pI73R Reveals GNB1 Binding and Host Gene Modulation

**DOI:** 10.3390/ijms262411768

**Published:** 2025-12-05

**Authors:** Katarzyna Magdalena Dolata, Barbara Bettin, Richard Küchler, Katrin Pannhorst, Dmitry S. Ushakov, Walter Fuchs, Axel Karger

**Affiliations:** Institute of Molecular Virology and Cell Biology, Friedrich-Loeffler-Institut, Federal Research Institute for Animal Health, Südufer 10, 17493 Greifswald-Insel Riems, Germany

**Keywords:** African swine fever virus, ASFV, pI73R, virus–host interaction, protein–protein interaction, GNB1, gene expression, CRNKL1

## Abstract

African swine fever virus (ASFV) causes a highly fatal disease in domestic pigs, resulting in substantial economic losses to the global swine industry. Vaccine development continues to be hindered by limited characterization of viral proteins and their functional redundancies. In this study, we employ combined experimental and computational approaches to characterize the ASFV I73R protein (pI73R), which contains a Z-DNA binding domain and plays a critical role in ASFV virulence and pathogenesis. We demonstrate that pI73R shares significant structural similarity with transcription factors of the forkhead box (FOX) protein family. Overexpression of pI73R results in downregulation of Crooked neck-like protein 1 (CRNKL1), a core spliceosome component, suggesting a potential mechanism by which pI73R modulates host protein synthesis. Using high-resolution mass spectrometry, we map the pI73R interactome and identify the host protein Guanine nucleotide-binding protein subunit beta-1 (GNB1) as a novel direct interactor of pI73R which may facilitate its nuclear transport. Furthermore, we show that pI73R exhibits consistent oligomerization and expression across different ASFV genotypes, highlighting its functional importance. Taken together, these results provide new insights into pI73R function, ASFV–host dynamics, and offer promising directions for antiviral strategy development.

## 1. Introduction

African swine fever (ASF) is a contagious and often lethal disease that affects both domestic pigs and wild boars, with lethality rates often approaching 100% depending on the strain. Due to its severe impact on animal health and agriculture, ASF is classified as a notifiable disease by the World Organisation for Animal Health.

The causative agent, African swine fever virus (ASFV), is a large double-stranded DNA virus from the *Asfarviridae* family, with a genome size ranging from 170 to 193 kb [1]. ASFV shares some genomic DNA and virion structure similarities with other large DNA viruses infecting eukaryotes, including members of the *Poxviridae* and *Iridoviridae* families. The virus primarily replicates in porcine macrophages, but it can also infect other cell types, especially in the later stages of the disease [2,3]. ASFV is endemic to sub-Saharan Africa, where it was initially discovered [4], and where 24 different genotypes, defined by the C-terminal sequence of the major capsid protein gene *B646L*, are present. A highly virulent genotype II virus is predominantly responsible for current outbreaks in domestic pigs and wild boar across Europe, Asia, and the Caribbean, causing significant economic losses in the swine industry [5].

ASFV encodes over 150 proteins involved in multiple stages of the viral lifecycle, including entry, replication, assembly, release, and the manipulation of host cell functions like immune evasion and modulation of protein expression. However, more than half of ASFV proteins remain functionally uncharacterized. Modified ASFV strains with deletions of virulence-associated genes have emerged as promising vaccine candidates, and were shown to confer protection against infection with homologous virulent viruses (reviewed in [6]). Nevertheless, aside from a vaccine licensed in Vietnam [7,8] no commercial ASF vaccine is available, largely due to the virus’s complexity and the limited understanding of its genetics and the immune reactions it triggers. Ongoing research into ASFV gene functions is essential to overcome these challenges and advance vaccine development.

The ASFV protein I73R (pI73R) contains a Z-DNA binding domain (Zα) [9] known to specifically bind left-handed conformation of nucleic acids including dsDNA, DNA/RNA hybrids, and dsRNA [10]. During ASFV infection, pI73R exhibits a preference for binding host cellular RNAs with high GC content [11]. Initially localized to the nucleus, pI73R gradually translocates to the cytoplasm as infection progresses [9]. Although pI73R is not essential for viral replication, it is crucial for effective virus production in vitro. Deletion of *I73R* leads to a complete loss of virulence and pathogenicity, making the deletion mutant a potential candidate for a live-attenuated vaccine [11]. pI73R also inhibits host protein synthesis [11,12] and promotes the nuclear retention of TNF-α mRNA [11]. Furthermore, independent of its Z-DNA binding activity, pI73R suppresses the cGAS-STING pathway’s induction of IFN-β [13]. Despite these insights, the exact mechanism by which pI73R suppresses host protein synthesis and innate immune responses remains unclear.

In this study, we demonstrate that pI73R exhibits structural similarity to host DNA- and RNA-binding proteins, particularly transcription factors of the forkhead box (FOX) family. Furthermore, the significant downregulation of crooked neck-like protein 1 (CRNKL1) in response to pI73R expression suggests a potential mechanism by which pI73R suppresses host protein synthesis. We also identified a novel direct interaction between pI73R and the guanine nucleotide-binding protein subunit beta-like protein 1 (GNB1), which may facilitate the nuclear transport of pI73R during infection. Collectively, these findings provide new insights into the role of pI73R in modulating host gene expression during ASFV infection.

## 2. Results

### 2.1. pI73R Protein Feature Analysis

The sequence of the genotype II ASFV (Georgia 2007/1) pI73R (GeneBank ID: CAD2068501.1) was analyzed for conserved domains (CDD), occurrence of signal peptides (SignalP), transmembrane domains (TMHMM), and disordered regions (IUPRED) without significant results. Also, a sequence homology search of the Uniprot/Swissprot database with NCBI BLAST did not produce significant hits outside the ASFV species (taxonomy ID 10497). Prediction of phosphorylation sites with GPS6.0 indicated possible phosphorylations of S8, Y17, and Y64 with scores over the cutoff. S8 was also the highest-ranking phosphorylation site predicted with Netphos 3.1, which also indicated possible phosphorylations at T3, Y17, T22, T40, Y48, S49, S51, and T62, but not at Y64. The Musite webtool indicated methylation of K5 and ubiquitinylation of K11. Protein structure-based homology prediction of the pI73R structure (PDB: 7VWV) performed with Foldseek search yielded 1991 significant (e-value ≤ 0.01) unique hits in the five available databases (Appendix A). Of these, 111 hits could be mapped to 58 unique human genes in the Uniprot database. Fourty genes were from the FOX family, and 46 were annotated in the Interpro database to belong to the Winged helix-like DNA-binding domain superfamily (IPR036388). Significant hits in the CATH database [14] suggested that pI73R has structural similarity to the CATH Superfamily 1.10.10.10 (Winged helix-like DNA-binding domain superfamily/Winged helix DNA-binding domain, Interpro IPR036388) [15]. This domain is known to recognize various nucleic acid structural features and is commonly utilized by transcription factors for sequence-specific DNA binding [16]. Accordingly, a functional analysis performed with g:Profiler [17] on 58 human structural homologues of pI73R revealed significant enrichment of gene ontology (GO) terms [18] associated with DNA-templated transcription (GO:0006351) and double-stranded DNA binding (GO:0003690). Pathway analysis with the Kyoto Encyclopedia of Genes and Genomes (KEGG) [19] and Reactome [20] further indicated an overrepresentation of pathways involved in the regulation of gene expression by FOX family proteins (KEGG:04068; REAC:R-HSA-9617828) (Figure 1, Appendix A). Interestingly, the enrichment results from the TRANSFAC (TF) database [21] indicate an overrepresentation of binding sites matching the specific DNA sequence motif *NTTGGCGGATGM*, which is recognized by the transcription factor ZNF286 (TF:M05958).

### 2.2. pI73R Homology with FOX and Z-DNA-Binding Proteins

Previously, Sun et al. [9] demonstrated that pI73R shares a high degree of structural similarity with several Z-DNA-binding proteins, including human ADAR1 (PDB: 1QBJ), goldfish PKZ (PDB: 4KMF), mouse DAI (PDB: 1J75), Yaba monkey poxvirus E3L (PDB: 1SFU) and Cyprinid herpesvirus 3 ORF112Zα (PDB: 4HOB). In the present study, structural comparison using Foldseek [22] identified an additional Z-DNA-binding protein, human ZBP1 (PDB: 2LNB), which exhibits significant structural similarity to pI73R. Furthermore, Foldseek identified notable structural overlaps with multiple members of the human FOX protein family. To quantitatively assess the degree of structural similarity between pI73R, Z-DNA-binding proteins, and FOX family proteins, we performed pairwise structural alignments using the TM-align server [23]. The template modeling (TM) scores for each protein pairwise comparison are presented in Table 1. These results indicate that both protein groups, Z-DNA binding and FOX family proteins, exhibit statistically significant structural alignments and a high degree of topological similarity with pI73R (TM score > 0.5). Additionally, we compared the amino acid sequence of pI73R with those of selected FOX proteins. Despite the low overall sequence identity, several conserved residues were identified that were shared between the FOX and the pI73R proteins of the genotype I and II strains BA71, and Georgia 2007/1, respectively (Figure 2). Strongly conserved positions were dominated by hydrophobic residues (L/I in alignment positions 10, 11, 25, 31, 67 and F in position 77) but also occurred for uncharged hydrophilic amino acid S (positions 66 and 72) and for charged amino acids (K in positions 81 and 84).

### 2.3. pI73R Is a Highly Abundant Early ASFV Protein with Genotype II-Specific Amino Acid Variation

As shown previously [11], pI73R is a highly conserved protein with only a few mutations identified between ASFV isolates. Interestingly, a comparison of the pI73R sequence across isolates shows a distinct substitution of tyrosine (Y) with histidine (H) in the genotype II variant. Additionally, genotype II lacks a threonine (T) at the C-terminus, which is present in genotypes I, X, and IX (Figure 3). According to AlphaFold protein structure modeling, these amino acid changes do not appear to affect the overall structural conformation of the protein (Figure 4). However, to further assess if the mutations affect pI73R characteristics and expression profile, we generated a polyclonal antibody serum against pI73R. We infected wild boar lung-derived (WSL) cells with ASFV strains Armenia/07 and Kenya1033, representing genotypes II and IX, respectively. With our antiserum we detected band patterns strongly suggesting the formation of oligomers, especially dimers, and trimers of pI73R in WSL cells infected with both ASFV genotypes (Figure 5A). Moreover, time-course immunoblot analysis showed that pI73R expression was first detectable at 8 h post-infection (hpi) in cells infected with either genotype II or IX of ASFV (Figure 5B). We employed MS to profile the temporal expression of viral proteins, including pI73R, throughout the infection. Samples were collected at 2, 4, 8, 12, 16, 18 and 24 h post-infection. The MS analysis confirmed the expression of pI73R beginning at 8 h post-infection. Furthermore, pI73R ranked among the top 10 most abundant viral proteins between 8- and 18 hpi (Figure 5C, Appendix A).

### 2.4. pI73R Overexpression Induces CRNKL1 Downregulation

To investigate the functional role of pI73R in regulating cellular processes, we analyzed changes in the cell proteome profile resulting from its overexpression. WSL cells were transfected with either an pI73R-GFP or a GFP construct as a control. After 24 h, cell lysates were collected from three independent replicates. Proteins were extracted and digested into peptides, which were then analyzed via MS for proteomic identification. The analysis identified a total of 4629 proteins (Appendix A), with 70 proteins significantly upregulated and 21 proteins downregulated in response to pI73R overexpression (Figure 6A). Among the downregulated proteins, CRNKL1 showed the most significant decrease, with a log2 fold change of −6.2. Additionally, the expression of pI73R led to a downregulation of several proteins involved in innate immune responses, including B-cell lymphoma/leukemia 10 (BCL10) and NF-kappa-B essential modulator (IκBKG or NEMO), which are key regulators of NF-κB signaling, as well as MAP kinase-activated protein kinase 2 (MAPKAPK2), a component of the MAPK14/p38α signaling pathway. To gather insights into the cellular function of CRNKL1, we analyzed its protein interaction network using the IntAct [24] online database. A total of 95 experimentally identified interactions between CRNKL1 and human proteins were identified (Appendix A). The GO-term and the KEGG pathway enrichment analysis of these interacting proteins was performed using g:Profiler and revealed a significant association of CRNKL1 with the mRNA splicing proteins (Figure 6B, Appendix A). CRNKL1 interacts with spliceosomal proteins of U2 and U4/U6-U5 small nuclear RNA (snRNA) complexes, PRP19-CDC5L complex, and the intron binding complex (Figure 6C). Additionally, several CRNKL1-interacting proteins have been reported to enhance or inhibit viral replication (Appendix A).

### 2.5. pI73R Protein Interaction Network

As a second approach to investigate the pI73R function, we analyzed the pI73R interactome in non-infected and ASFV-infected host cells. WSL cells were transfected with a GFP or an pI73R-GFP fusion construct for 24 h. Subsequently, affinity purification- mass spectrometry (AP-MS) was employed to identify interacting proteins, following the procedures previously described [25] (Figure 7A). The expression of pI73R in host cells was confirmed by immunoblot (Appendix A) and MS (Appendix A). We identified 236 host proteins that specifically coprecipitated with pI73R in non-infected WSL cells (Appendix A). The GO and KEGG terms enrichment analysis of pI73R-coprecipitated proteins (Appendix A) showed significant enrichment in host proteins related to translation (GO:0006412), ribosome (GO:0005840) and RNA splicing (GO:0008380) (Figure 7B). To enhance the clarity of the functional landscape of the pI73R interactome, we cataloged proteins within the network based on their experimentally validated interactions obtained from STRING database [5]. We then employed the clusterMaker2 Cytoscape application to perform Markov Clustering (MCL) to cluster these proteins into functional groups. The results of the clustering were consistent with our enrichment analysis, revealing two prominent clusters: one comprising proteins involved in cytoplasmic translation and the other associated with RNA splicing (Figure 7C). The latter cluster included proteins that also interact with CRNKL1, such as HNRNPC, SRSF3, SRSF6, SRSF7, SNRPE, and ILF2. Moreover, aside from ribosomal proteins, the most abundant proteins associated with pI73R were the guanine nucleotide-binding proteins G(I)/G(S)/G(O) subunit gamma-2 (GNG2) and beta-1 (GNB1), which are part of the G protein-coupled receptor (GPCR) signaling pathway cluster.

### 2.6. pI73R Interacts with GNB1 Protein in ASFV-Infected Cells

To identify the molecular interactions mediated by pI73R during host cell infection we transfected WSL cells with the pI73R protein and subsequently infected them with the ASFV Armenia/07 for 18 h. Flow cytometry analysis revealed a high transfection efficiency, with over 90% of the cells successfully expressing the transgene (Appendix A). Next, we compared the pI73R interactomes in mock and ASFV-infected cells. Under infection conditions, the pI73R interactome was notably reduced, comprising 25 proteins (Appendix A), whereas, in mock-treated cells, it included 236 proteins (Appendix A). Despite this reduction, 21 interactors were common to both conditions, with GNB1 and GNG2 showing the highest levels of enrichment (Figure 8A). We proceeded to examine whether pI73R directly interacts with the GNB1 and GNG2 proteins. Structural modeling of the complexes between pI73R and the proteins GNB1 and GNG2 was performed using AlphaFold2, resulting in a high-confidence model of the 1:1 pI73R-GNB1 complex (Figure 8B). This interaction was experimentally validated using the far Western blot technique. In this assay, purified pI73R-GFP and GFP (as a control) were separated via gel electrophoresis and transferred onto a membrane. Following a transfer, the membrane was blocked and probed with purified GNB1-GST. After washing, GNB1 binding to pI73R, but not to GFP, was detected using a GNB1-specific antibody (Figure 8C). Both AlphaFold modeling and the far Western blot technique failed to detect any direct interaction between pI73R and GNG2. Additionally, confocal microscopy of WSL cells expressing GNB1 and pI73R showed that both proteins are broadly distributed throughout the cell, with only limited co-localization detectable in structures resembling the cytoskeleton (Appendix A).

## 3. Discussion

ASFV is a member of the Nucleocytoplasmic Large DNA Viruses (NCLDV) family, characterized by large genomes and replication being either cytoplasmic or requiring both cytoplasmic and nuclear stages [26]. ASFV replication mainly occurs in cytoplasmic viral factories, beginning at approximately 4–6 hpi [27,28]. Several studies reported the presence of viral DNA also in the host nucleus [27,29,30] and suggest that a nuclear phase is essential for successful infection [31]. However, the recent study by Weng et al. [28] provides strong evidence that ASFV DNA does not involve a nuclear stage. These conflicting findings highlight the need for further research to clarify the early events of ASFV replication and the functions of early-expressed viral proteins.

Within the initial four hours of infection, ASFV induces several nuclear alterations, including the formation of nuclear blebs [29] and the triggering of the host DNA damage response [32]. It also disrupts the nuclear lamina, releasing nuclear material into the cytoplasm [33], and influences chromatin remodeling [34]. Collectively, these changes modify the nuclear environment to promote conditions favorable for ASFV replication. Despite the recognized importance of the nuclear phase in ASFV replication, the mechanisms by which the virus manipulates nuclear processes remain largely uncharacterized, and no specific cellular or viral proteins have been implicated in this process.

Similar to other large DNA viruses like poxviruses and herpesviruses, ASFV utilizes a temporal gene expression strategy during its replication cycle. Early expressed genes are crucial for establishing viral replication by inhibiting host immune defenses and promoting viral DNA synthesis, while late genes are mostly involved in producing structural components required for assembling and releasing new virions. Investigating the roles of both structural and early-expressed genes, particularly those that localize to the nucleus, could provide critical insights into the molecular mechanisms driving the initial nuclear phase of infection and the involvement of host proteins in this process.

In this study, we integrated computational and experimental approaches to investigate the structural and functional properties of the pI73R protein, a viral component known to be both structural [35] and early-expressed [11,36] and to localize to the nucleus during infection [9]. We report several findings that reveal previously unknown characteristics of the protein, offering new insights into its role during infection.

Firstly, the results of computational analysis of protein structure homology strongly suggested that the pI73R protein has DNA- and/or RNA-binding ability, which was not detected by sequence-based structural domain search. We showed that, in addition to Z-DNA-binding proteins [9], pI73R also structurally resembles transcription factors of the FOX family (Figure 2). FOX proteins play important roles in maintaining cell homeostasis and engage in regulating the expression of multiple immune-related genes [37,38,39]. Moreover, cellular Z-DNA-binding homologues of pI73R, such as ZBP1, PKZ, and ADAR1, are involved in antiviral interferon response pathways and the inhibition of viral replication [40,41,42,43], while viral homologues, including the E3L protein of vaccinia virus and ORF112 of cyprinid herpesvirus 3, have been implicated in viral mimicry by acting as dsRNA-binding proteins that suppress the interferon response [44,45]. In this context, the interesting question is whether pI73R could suppress protein synthesis [11,12] by functionally mimicking transcription factors, thereby inhibiting the expression of specific host genes. Furthermore, given its structural similarity to viral E3L and ORF112 proteins, it is worth considering whether pI73R might also suppress interferon responses [13] through a comparable mechanism.

To approach this questions experimentally, we transfected I73R into swine cells and conducted a whole-proteome analysis to identify specific host proteins affected by pI73R overexpression. Our results revealed a significant reduction, by more than 6 log2 fold, of the levels of the host protein CRNKL1 in response to pI73R expression in non-infected cells (Figure 6A). CRNKL1, a core spliceosome component, plays a critical role in pre-mRNA splicing [46]. Additionally, CRNKL1 associates with the Prp19 complex, which is involved in various cellular processes, including splicing, transcription, mRNA export, genome maintenance, protein degradation, and lipid droplet biogenesis (reviewed in [47]). Interestingly, the Prp19 complex plays also a role in nuclear export of RNA by interacting with the TREX export machinery early in RNA synthesis, potentially determining whether an RNA transcript is targeted for export or retention [48]. CRNKL1 can directly bind RNA via its half-a-tetratricopeptide repeat (HAT) motif [49] and the knockdown of CRNKL1 results in nuclear retention of the naturally intronless mRNAs [48].

For human immunodeficiency virus type-1 (HIV-1), CRNKL1 is a critical host factor that negatively regulates its complex splicing pattern. Xiao et al. [50] demonstrated that CRNKL1 binds to RNA motifs within specific subsets of cellular and viral introns, and its depletion leads to increased expression and nuclear export efficiency of unspliced HIV-1 RNA. This unspliced RNA is essential for the synthesis of structural viral proteins and serves as the viral genome for packaging. Moreover, CRNKL1 knockdown selectively decreases cytoplasmic levels of host mRNAs involved in cell cycle regulation and organelle organization [50], potentially impacting these host cell functions. To the best of our knowledge, there is currently no evidence that ASFV produces spliced transcripts [51]. Therefore, pI73R-mediated suppression of CRNKL1 is more likely to interfere with host mRNA splicing and/or nuclear export, reducing the cytoplasmic availability of specific host factors that could otherwise limit ASFV replication. Further studies are needed to confirm CRNKL1 downregulation by I73R during ASFV infection, and to clarify the underlying mechanism. It will also be important to determine whether the observed I73R-mediated host shutoff [11,12] results from the nuclear retention of spliced host mRNAs during early infection due to CRNKL1 depletion.

It is noteworthy to mention here that, unlike Liu and colleagues [11], we did not detect a significant suppression of host gene expression in porcine cells overexpressing pI73R. However, comprehensive and accurate monitoring of pI73R-induced shutoff was beyond the scope of this study and would require MS analysis combined with isotope labeling as the label-free quantitation approach used in this study does not capture protein synthesis with high sensitivity [52].

A second important observation of this study resulted from comparing pI73R sequences. Liu and colleagues [11] previously reported that I73R is highly conserved across ASFV strains, with only minor sequence variation. Consistent with those observations, we noted a specific variation between the currently circulating pandemic genotype II strain, and genotypes I and IX (Figure 3). This raised the question of whether the pI73R sequence differences affect its oligomerization ability, expression levels, or temporal dynamics during viral infection.

Independent transcriptomic and proteomic studies across various cell lines infected with different ASFV isolates have consistently identified pI73R as one of the most highly expressed genes and proteins during ASFV infection. Namely, in their transcriptomic analysis, Cackett et al. [36] reported high pI73R expression levels at 5 and 16 hpi in Vero cells infected with the ASFV BA71V strain (genotype I). Additionally, proteomic studies by Kessler et al. [53] in WSL, Vero, and HEK293 cells infected with ASFV OURT 88/3 (genotype I), and by Wöhnke et al. [54] in macrophages infected with ASFV-Kenya1033-ΔCD2v-dsRed (genotype IX), confirmed that pI73R remains one of the most abundant proteins at 24 hpi and 48 hpi, respectively. Consistent with previous transcriptomic and proteomic studies, we confirmed that also the genotype II variant of pI73R is expressed early (8 hpi) and remains highly abundant throughout infection (Figure 5B). Furthermore, at early stages of infection, pI73R is abundantly expressed alongside other viral proteins, such as MGF110-7L, I215L, and CP312R, that inhibit host protein translation [55,56,57].

Moreover, the sequence variation between pI73R from genotypes II and IX does not appear to affect its overall structure (Figure 4), ability to form multimers (Figure 5A), and protein expression pattern during infection (Figure 5B). However, the tyrosine-to-histidine substitution in the genotype II variant could still influence protein–ligand interactions. The imidazole ring and positive charge of histidine provide a higher affinity than tyrosine for interacting with the negatively charged phosphate backbone of DNA or for coordinating metal ions (e.g., Zn^2+^) that help to stabilize DNA-binding sites [58]. Also, the predicted post-translational modifications (PTMs), indicated by multiple phosphorylation sites, one methylation, and one ubiquitination site, suggest that pI73R may undergo dynamic structural changes during infection that could modulate its nuclear localization, protein–protein interactions, DNA-binding affinity, stability, and oligomerization. However, neither of the deleted residues (Y34 or T72) was predicted to be phosphorylated by any of the webtools used in this study.

Also, we confirmed that pI73R can form stable di-, and trimers in infected cells, as it does in non-infected cells [9]. Western blots in Figure 5A indicate the formation of additional higher-order homo- or heterooligomers which remain to be analyzed in detail.

Finally, the function of the pI73R protein has previously been examined only in non-infected cells [9,13]. However, since viral infection profoundly alters the cellular environment—potentially impacting protein structure, localization, and interaction partners—we sought to compare the interactome of pI73R in non-infected versus ASFV-infected cells. Our results revealed a large discrepancy in the number of host proteins interacting with pI73R in non-infected versus ASFV-infected cells (Figure 8A). Notably, the translation-related and spliceosomal proteins, which interact with pI73R in non-infected cells (Figure 7B,C), were absent from the interactome during infection. The association of spliceosomal and ribosomal proteins with the pI73R interactome is consistent with previous findings that pI73R binds host RNAs and translocates between the nucleus, cytoplasm, and viral factories [11], positioning it near these host proteins. Although the absence of these interactions during ASFV infection appears surprising at first and requires experimental clarification, an explanation could be that the function of pI73R and consequently its interactome may change over the course of infection. In the early stages of infection, pI73R may modulate host and viral protein synthesis while localized in the nucleus. As the infection progresses and pI73R translocates to the cytoplasm and viral factories, its function may shift. At later stages, pI73R oligomers may be incorporated into viral particles, consistent with previous findings showing its presence in the virion [35]. At the late infection time point analyzed in this study, pI73R was observed in both the nucleus and cytoplasm of infected and uninfected cells. However, ASFV-induced nuclear disruption may account for the absence of nuclear protein interaction partners in infected cells. Additional experiments at earlier infection stages may help explain the striking differences in the pI73R interactome between uninfected and infected cells.

Despite these limitations, we identified the host protein GNB1 as a high-confidence interactor of pI73R in both conditions, and we subsequently confirmed that this interaction is direct (Figure 8B,C). GNB1 is a beta subunit of guanine nucleotide-binding proteins (G proteins), which play a key role in cell signaling by mediating the exchange of signals between cell surface receptors, such as GPCRs, and various intracellular effectors [59]. Although the role of G proteins in the viral life cycle is largely understudied, GNB1 has been identified as a key player in processes like nuclear import [60] and virus budding [61], specifically for the Influenza A virus (IAV). Especially intriguing is the role of GNB1 in the nuclear transport of the IAV polymerase subunit PB2. Research by Zheng et al. [60] demonstrated that GNB1 facilitates the binding between PB2 and importin alpha, promoting nuclear import of PB2 through the classical nuclear import pathway. Given that the ASFV protein pI73R lacks a nuclear localization signal (NLS), it may similarly “hitchhike” on host proteins with NLS capabilities to enter the nucleus. Although small proteins like pI73R can potentially diffuse passively through nuclear pores, we hypothesize that, as in IAV, GNB1 may support pI73R’s interaction with importins or other nuclear transport proteins to facilitate or enhance nuclear entry. pI73R, a virulence-associated protein essential for ASFV pathogenesis, could rely on GNB1 for effective nuclear transport, highlighting the potential of GNB1 as a target in antiviral strategies. Future studies will investigate the specific mechanism underlying the GNB1-pI73R interaction and GNB1′s broader role in ASFV replication.

## 4. Materials and Methods

### 4.1. Cells and Viruses

The WSL cells [62] were supplied by the Friedrich-Loeffler-Institut Biobank (catalog number CCLV-RIE 0379). Cells were maintained in Iscove’s modified Dulbecco’s medium (IMDM) mixed with Ham’s F-12 nutrient mix (1:1 [*v*/*v*]) supplemented with 10% fetal bovine serum (FBS). For infection experiments, the ASFV Armenia/08 isolate (genotype II), which is almost identical to ASFV Georgia 2007, was used after adaption to WSL cells by serial passaging. Passage 20 stocks were generated as described previously [63]. A WSL-adapted ASFV Georgia 2007 isolate was not available, and the wild-type Armenia and Georgia strains do not replicate in these cells. ASFV-Kenya1033-ΔCD2v-dsRed, derived from the genotype IX isolate ASFV-Kenya1033 (February 2013), expresses dsRed under the late p72 promoter in the CD2v locus. Its generation and partial characterization have been described previously [63].

### 4.2. Plasmids and DNA Transfection

For the generation of a plasmid expressing pI73R of ASFV Georgia 2007/1 (GenBank accession FR682468.2; encoded by nt. 173,088 to 173,306) fused to GFP, the vector pEGFP-N1 (Clontech, GenBank accession U55762) was first doubly digested with restriction enzymes BglII and BamHI and religated to remove noncoding sequences upstream of the GFP open reading frame (ORF). The resulting vector pEGFP-N1BB was linearized with BsrGI, and treated with DNA Polymerase I, Large (Klenow) Fragment to generate blunt ends. Subsequently, the purified vector was combined with the synthetic codon-adapted viral ORF pI73R flanked by 19 vector-overlapping nucleotides at both ends (purchased as Gene Strand from Eurofins Genomics (Louisville, KY, USA), Appendix A), and used for Hot-Fusion cloning [64]. In the resulting plasmid pEGFP-pI73Rporc, the synthetic pI73R ORF was under the control of the human cytomegalovirus (HCMV) immediate-early promoter/enhancer complex and 5′-terminally fused to the coding sequence of GFP. The accuracy of the resulting plasmid was confirmed by sequencing.

For the AP-MS, WSL cells were transiently transfected with either an pI73R-GFP or GFP vector as baits using the K2 multiplier and K2 transfection reagent (Biontex, Munich, Germany) according to the manufacturer’s protocol. Six hours post-transfection, the cells were infected with ASFV. For each bait, three independent biological replicates were prepared for MS.

### 4.3. ASFV Infection

All ASFV experiments were conducted in a biocontainment facility fulfilling the safety standards for ASF laboratories and animal facilities (Commission Decision 2003/422/EC, Chapter VIII). WSL cell monolayers were infected with ASFV stock dilutions at a multiplicity of infection (MOI) of 1 PFU/cell. Supernatants from uninfected cells were used as mock-infected controls. Following inoculation, cells were centrifuged at 600× *g* for 1 h at 37 °C. The cells were then washed three times with phosphate-buffered saline (PBS), replenished with medium containing 5% FBS, and incubated at 37 °C with 5% CO_2_ for the required duration.

### 4.4. Antibodies and Recombinant Proteins

The primary antibodies used for immunoblotting included rabbit anti-GFP (PABG1, Chromotek, Planegg, Germany), and mouse anti-GNB1 (sc-515764, SantaCruz, CA, USA). The secondary antibodies used were peroxidase-conjugated goat anti-mouse and anti-rabbit IgG (Jackson ImmunoResearch, Ely, UK). Recombinant human GST-tagged GNB1 protein (H00002782-P01) was purchased from Abnova (Taipei City, Taiwan).

### 4.5. Generation of Polyclonal Antiserum Against pI73R

The pI73R coding sequence from ASFV Georgia 2007/1 was cloned into the pGEX-4T3 plasmid, encoding an N-terminal GST tag and a thrombin cleavage site (Appendix A). The construct was expressed in *E. coli* XL1-Blue. Cells were grown in standard LB medium to an OD_600_ of 1.5–2.0. Bacteria were pelleted by centrifugation at 4000× *g* for 15 min at 4 °C, washed with PBS, and lysed in buffer containing 50 mM Tris-HCl (pH 8.0), 2.5 mM dithiothreitol (DTT), and 1 mM phenylmethylsulfonyl fluoride (PMSF). The clarified lysate was applied to a GSTrap™ High Performance column (#175281-01, Cytiva, Marlborough, MA, USA) according to the manufacturer’s protocol. On-column cleavage was performed overnight at 4 °C using 80 U thrombin protease (Cytiva, #10218854). Cleaved pI73R was separated from thrombin using a HiTrap^®^ Benzamidine Fast Flow column (Cytiva, #175143-02), following the manufacturer’s instructions. The flow-through, containing 100 µg of purified pI73R free of GST and thrombin, was used to immunize rabbits for polyclonal antiserum production. The resulting antibodies were isolated from rabbit sera and assessed for specificity.

### 4.6. Purification of pI73R-GFP and GFP for Western Blotting

WSL cells expressing pI73R-GFP or GFP were cultured in T75 flasks (Corning, Corning, NY, USA), harvested, and lysed with 5 mL of lysis buffer (50 mM Tris–HCl, pH 7.5, 150 mM NaCl, 1% NP-40, 10 mM DTT, 1 mM EDTA, and the cOmplete^TM^ mini EDTA-free protease inhibitor cocktail [no. 04693159001, Roche, Mannheim, Germany]). The cell lysates were incubated with 100 µL of GFP-Trap agarose beads at 4 °C for 1 h to allow binding. Following incubation, the agarose beads were collected using Poly-Prep columns by gravity flow. The beads were sequentially washed with 10 mL of wash buffer I (50 mM Tris–HCl, pH 7.5, 150 mM NaCl) and 10 mL of wash buffer II (50 mM Tris–HCl, pH 7.5, 300 mM NaCl) to remove non-specifically bound proteins. To eliminate heat shock proteins, the columns were additionally washed with 10 mL of wash buffer III (50 mM Tris–HCl, pH 7.5, 150 mM NaCl, 5 mM MgCl_2_, 1 mM ATP). Protein elution was performed by mixing the agarose with 1 mL of elution buffer (100 mM Glycine, pH 2.5) for 1 min. The eluates were immediately neutralized with 0.2 mL of 1 M Tris, pH 10.5. Next, purified proteins were resolved on 4 to 20% Mini-Protean TGX SDS-PAGE gels (Bio-Rad, Feldkirchen, Germany) [65] and transferred to nitrocellulose membranes using a Trans-Blot Turbo semi-dry transfer system (Bio-Rad Laboratories, Feldkirchen, Germany) [66]. The membranes were blocked with 5% non-fat dry milk in Tris-buffered saline with 0.25% Tween 20 (TBST) for 1 h and optionally incubated overnight at 4 °C with a GST-tagged GNB1 (10 μg/mL). After washing with TBST, the membrane was incubated with an anit-GNB1 antibody (1:100) and an HRP-conjugated secondary antibody (1:10,000) for 1 h each. Antibody binding was visualized using the Clarity Western enhanced chemiluminescence (ECL) substrate (Bio-Rad), imaged on a C-DiGit blot scanner (LI-COR, Bad Homburg, Germany), and analyzed with Image Studio software (v.5.2).

### 4.7. Generation and Analysis of MS-Sample

#### 4.7.1. Affinity Purification

For each affinity purification experiment 5 × 10^6^ cells were seeded. After an overnight incubation, the cells were transiently transfected for 6 h before ASFV infection. At 18 hpi and 24 h post-transfection, cells were washed with phosphate-buffered saline (PBS), lysed, and subjected to affinity purification using 50 μL of GFP-Trap agarose beads (Chromotek, Planegg, Germany), as previously described [25].

#### 4.7.2. On-Bead Digestion

Bead-bound proteins were suspended in 300 μL freshly made UA buffer (8 M urea, 100 mM Tris-HCl, pH 8.0), and loaded onto 10 kDa filter units (Sartorius, Göttingen, Germany). The samples were centrifuged at 12,000× *g* for 30 min at 20 °C. Filter-aided sample preparation (FASP) was used for trypsin digestion, performed as described previously in [67]. Proteins were digested on the beads in 100 μL of digestion buffer (1 M urea, 50 mM Tris-HCl, pH 7.5, and 5 μg/mL trypsin (V5111, Promega, Madison, WI, USA)). Digestion was carried out overnight at 37 °C with continuous shaking. The following day, the peptide-containing supernatant was collected by ultrafiltration and acidified with formic acid to a final concentration of 1%. Peptides were desalted using C18 100-μL tips (Thermo Scientific, Darmstadt, Germany) per the manufacturer’s instructions, dried by vacuum centrifugation, and reconstituted in 20 μL of 0.1% formic acid before MS analysis.

#### 4.7.3. Lysate Sample Preparation

Confluent monolayers of WSL overexpressing GFP or pI73R-GFP were washed three times with PBS and lysed in 2% SDS in 0.1 M Tris-HCl (pH 8.0) for 10 min at 95 °C. Lysates were clarified by centrifugation (14,000× *g*, 10 min, RT), and the supernatants were collected. Aliquots containing 100 µg protein (determined by BCA assay) were prepared using the FASP protocol, as described in the previous section.

#### 4.7.4. MS Data Acquisition and Analysis

Samples were analyzed on a timsTOF Pro mass spectrometer coupled to a nanoElute nanoflow liquid chromatography system (Bruker, Bremen, Germany). Peptides were separated on a reversed-phase analytical column (10 cm × 75 μm i.d., Bruker 1866154) with a gradient elution of 2 to 95% mobile phase B over 65 min (2% to 4% solvent B for 0 to 1 min, 4% to 20% solvent B for 1 to 46 min, 20% to 32% solvent B for 46 to 60 min, 32% to 95% solvent B for 60 to 61 min, and 95% solvent B for 61 to 65 min) at a constant flow rate of 250 nL/min. The column temperature was maintained at 40 °C. MS analysis of eluting peptides was performed in ddaPASEF mode with a cycle time of 1.1 s, as recommended by the manufacturer. Proteomic data were searched against the ENSEMBL [68] Sus scrofa proteome (v.11.1.2021-11-10) and the NCBI ASFV Georgia proteome (version FR682468.2) using MaxQuant software (v.2.0.2.0) with default settings [69]. The ASFV Georgia proteome was used because it shares 99.99% genome sequence identity with the Armenia/08 isolate, which lacks a publicly available, reviewed proteome. The false discovery rates (FDR) for peptides and proteins were set at 1%, with a minimum peptide length of seven amino acids. The match-between-runs option was enabled with a 0.7 min match window for retention times 20 min alignment time, and a tolerance of 0.05 for 1/k_0_. Data were analyzed using Perseus software (v. 2.1.0.0) [70]. A protein was identified as interactor if at least two unique peptides were found in two out of three replicates. The GFP background included proteins detected in GFP negative controls. Potential interactors were considered specific if they were found only in pI73R-GFP pulldowns but not in the background or if the log2 fold change between GFP control and GFP-bait exceeded 2 with a *p*-value of <0.01 from a two-sided *t*-test.

### 4.8. Bioinformatics Analysis

#### 4.8.1. Term Enrichment Analysis

Porcine genes were assigned to their corresponding human orthologues using the R package gprofiler2 (v.0.2.1) [71]. The overrepresentation analysis was performed using the enricher function of the clusterProfiler (v.4.2.2) [72] package in R with default parameters. Significantly enriched GO [2] and KEGG [3] terms (adjusted *p* value  <  0.01) were identified and further clustered based on their semantic similarity using the R package rrvgo (v.1.6.0) [73].

#### 4.8.2. Protein–Protein Interaction Networks

Protein interaction data were imported into Cytoscape (version 3.9.1) [74] to construct a protein–protein interaction network. The ClusterMaker2 plugin [75] was utilized to perform functional clustering using the Markov Cluster Algorithm (MCL). Parameters for MCL were set with an inflation value of 2 to identify densely connected protein clusters.

#### 4.8.3. Computational Modeling of Proteins and Protein Complexes

Protein predictions of pI73R BA71V was generated using the ColabFold version 1.5.5 [76,77] webserver. A total of 5 models were predicted for BA71V pI73R with 3 recycles. Predictions were run in template mode using the pdb100 database and models were ranked by their pLDDT scores. Predictions of protein complexes were performed using the Alphafold-Multimere [78] module of ColabFold. A total of 5 models were predicted for each complex with 3 prediction cycles. Model relaxation and energy minimization was performed using the integrated Amber module. The confidence of resulting protein complex predictions was assessed based on the predicted aligned error (PAE) scores.

#### 4.8.4. Structure Analysis

The following web-based tools were used to analyse structural features of pI73R protein: CDD [79], Netphos [80], IUPred [81], SignalP [82], DeepTMHMM [83]. Potential phosphorylation sites were predicted with GPS 6.0 [84] and MusiteDeep [85], Structure-based searches were carried out with Foldseek [22]. Protein structure alignment score between pI73R and homologues proteins was calculated using TM Align (v.20190822) [23].

#### 4.8.5. Multi Sequence Alignment

The MSA was performed using the MAFFT version 7.511 webserver [86]. Default parameters were kept and the MSA was performed with the L-INS-I refinement method, with minor manual adjustments applied. The following UniProt FASTA sequences were used for alignment: I73R (Georgia) (A0A2X0RU36), I73R (BA71V) (P27946), FOXA1 (P55317), FOXH1 (O75593), FOXN3 (O00409), FOXO1 (Q12778), ZBP1 (Q9H171), and ORF112 (A4FTK7).

## 5. Conclusions

This study focuses on the characterization of the ASFV protein pI73R, a critical factor in viral virulence and pathogenesis and a potential vaccine target. Its significant homology to host FOX family transcription factors, together with the strong downregulation of CRNKL1 expression, suggests a mechanism by which pI73R may mediate host protein shutoff through nuclear retention of spliced host RNAs. Furthermore, we identified the host protein GNB1 as a novel and direct interactor of pI73R, which, based on its known characteristics, may facilitate pI73R’s nuclear import. These findings highlight new directions for investigating the mechanism of the pI73R–GNB1 interaction and the role of CRNKL1 in ASFV pathogenesis.

## Figures and Tables

**Figure 1 ijms-26-11768-f001:**
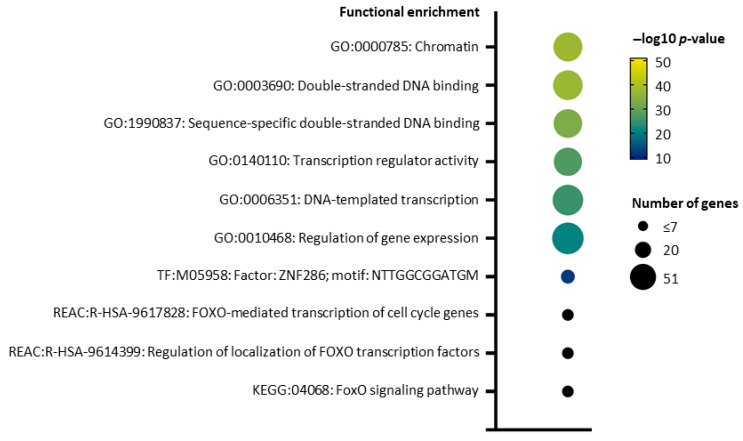
Selected top functional terms from overrepresentation analysis for 58 human structural homologues of pI73R identified with Foldseek search, highlighted in grey in Appendix A.

**Figure 2 ijms-26-11768-f002:**
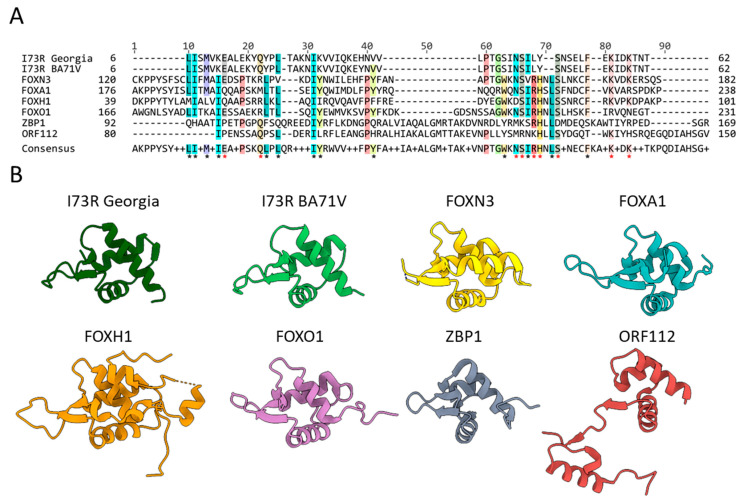
Overall structural and sequence comparison between pI73R and known Z-DNA interactors, including FOX family proteins. (**A**) Multi sequence alignment (MSA) of the pI73R amino acid sequence with those of structurally homologous FOX family proteins and Z-DNA interactors. Positions of conserved amino acids (≥50%) across sequences are highlighted and a MSA-based consensus is given below. (**B**) Aligned protein structures of I73R Georgia (PDB: 7VWV), I73R BA71V (prediction), FOXN3 (PDB: 6NCM), FOXA1 (PDB: 7VOX), FOXH1 (PDB: 7YZB), FOXO1 (PDB: 3CO7), ZBP1 (PDB: 3EYI), and ORF112 (PDB: 4HOB). Consensus positions with hydrophobic or hydrophilic amino acid residues are marked with black and red asterisks, respectively.

**Figure 3 ijms-26-11768-f003:**
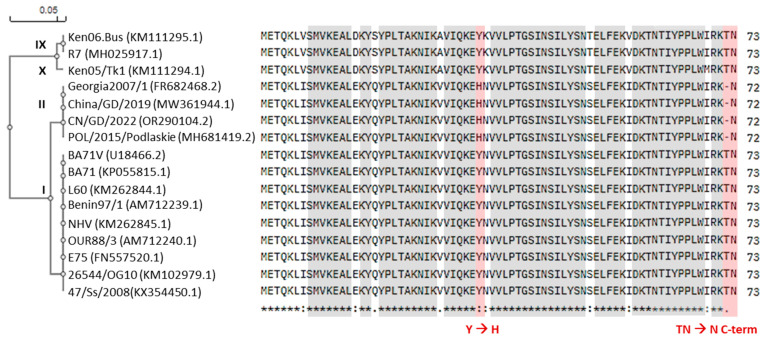
Alignment of pI73R sequences across different ASFV genotypes, highlighting conserved and variable regions. The consensus line below the alignment indicates identical residues (*), strongly conserved substitutions (:), and weakly conserved substitutions (.). Unique amino acid changes between the ASFV genotype II I73R sequence and the sequences from genotypes I, IX, and X are highlighted in red. Amino acid sequences were obtained from the NCBI GenBank database. Corresponding accession numbers are indicated.

**Figure 4 ijms-26-11768-f004:**
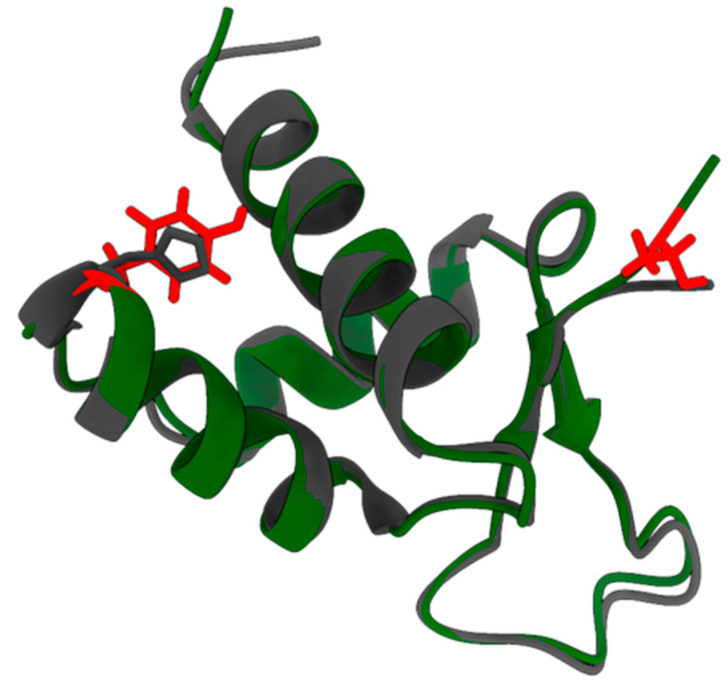
Overlay of the AlphaFold model of genotype II pI73R (PDB: 7VWV, shown in grey) and the predicted genotype I BA71V pI73R model (in green). Side chains of residues Y34 and T72 of the BA71V pI73R are highlighted in red. Side chain of the genotype II-specific H34 mutation is shown in grey.

**Figure 5 ijms-26-11768-f005:**
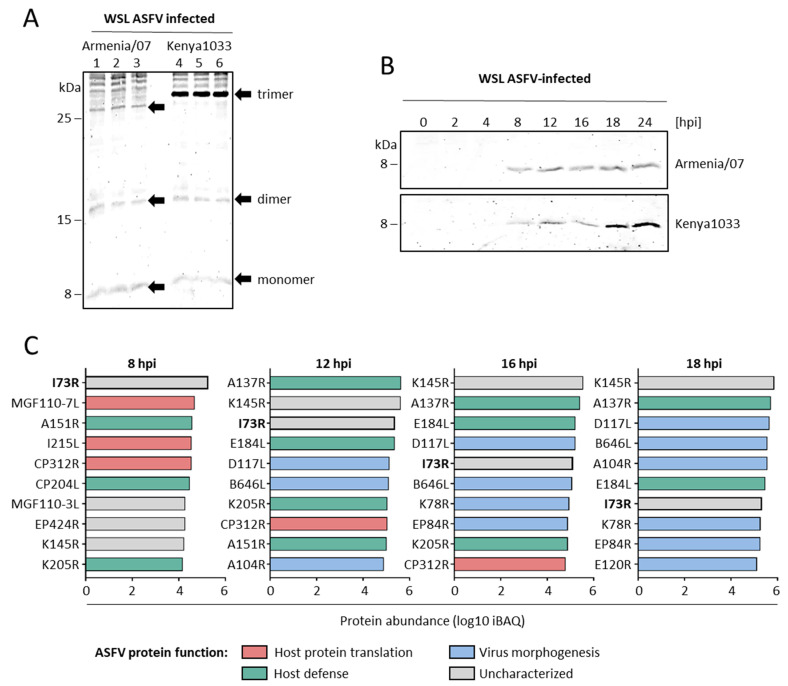
Characterization of pI73R oligomerization and expression profile. (**A**) Immunoblot analysis depicting the formation of pI73R oligomers in WSL cells infected with ASFV Armenia/07 (genotype II; triplicates 1–3) and Kenya1033 (genotype IX; triplicates 4–6) at an MOI of 1 PFU/mL for 24 h. (**B**) Time-course immunoblot analysis of pI73R expression in WSL cells infected with ASFV Armenia/07 and Kenya1033 at an MOI of 1 PFU/mL, with samples collected at 0, 2, 4, 8, 12, 16, 18 and 24 h post-infection (Appendix A). (**C**) Top 10 most abundant ASFV proteins ranked according to their expression levels determined by quantitative MS at the indicated times postinfection. Proteins were categorized based on their functional roles during infection.

**Figure 6 ijms-26-11768-f006:**
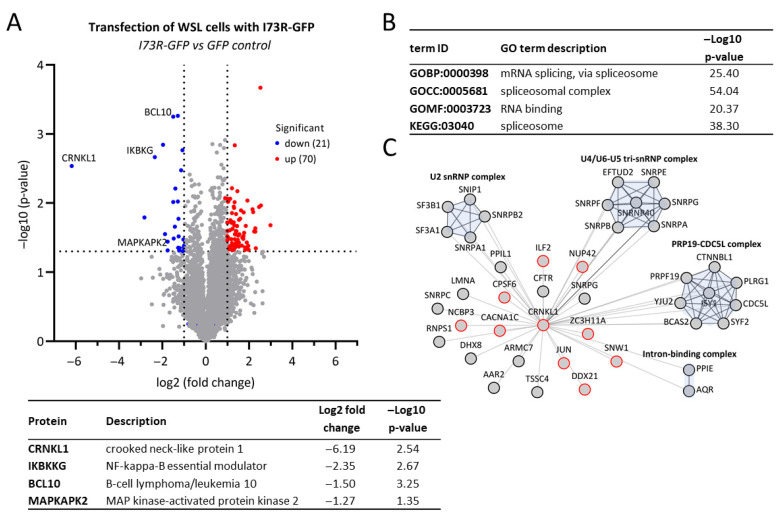
ASFV pI73R expression results in the downregulation of CRNKL1, a protein involved in mRNA splicing. (**A**) Volcano plot showing differentially expressed genes in WSL cells expressing pI73R-GFP or GFP control. The −log10 *p*-value (Benjamini–Hochberg corrected) is plotted against the log2 fold change (pI73R-GFP/GFP). The dotted vertical lines denote ±1.0-fold change on the log2 scale while the dotted horizontal line denotes the significance threshold of *p* = 0.05 (Appendix A). Red and blue dots indicate significantly up- and down-regulated proteins with at least twofold changes in abundance, respectively. (**B**) Top functional terms enriched among host proteins interacting with CRNKL1 from the IntAct database (Appendix A). (**C**) Network illustration of the CRNKL1 interactome highlighting spliceosomal proteins and proteins implicated in viral infection. Proteins were grouped into protein complexes according to the EBI Complex Portal. Proteins involved in virus replication are outlined in red and highlighted in grey in Appendix A.

**Figure 7 ijms-26-11768-f007:**
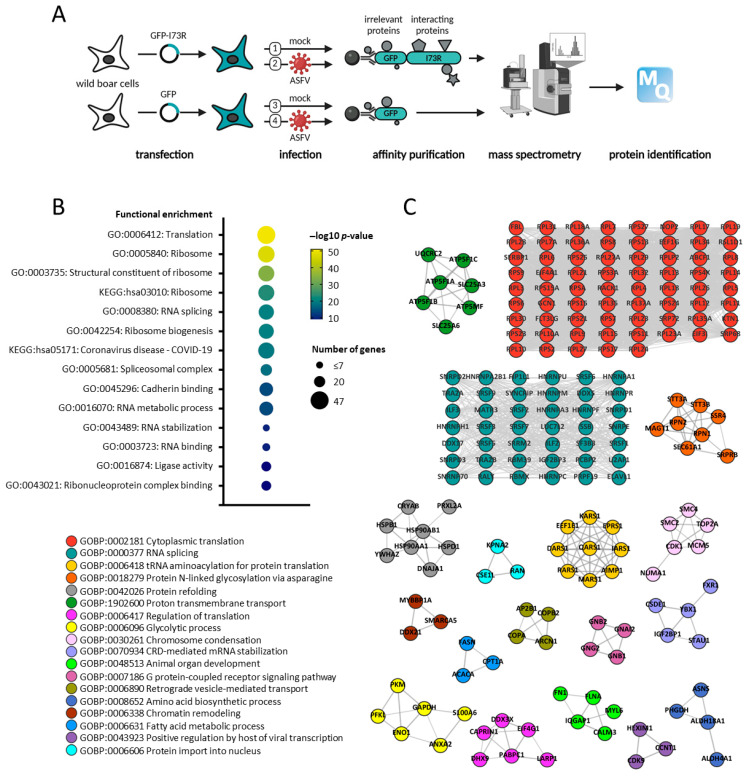
Constituents of the pI73R interactome. (**A**) AP-MS experimental workflow for identifying host and virus protein–protein interaction of pI73R (Appendix A). (**B**) Selected top functional GO and KEGG terms from overrepresentation analysis are shown for pI73R interactors in non-infected cells and are highlighted in grey in Appendix A. (**C**) Clusters of proteins generated by clusterMaker 2.0 Cytoscape application based on the pI73R protein–protein interaction network in non-infected WSL cells. Colors indicate biological pathways according to the Gene Ontology. Singletons were excluded from the network.

**Figure 8 ijms-26-11768-f008:**
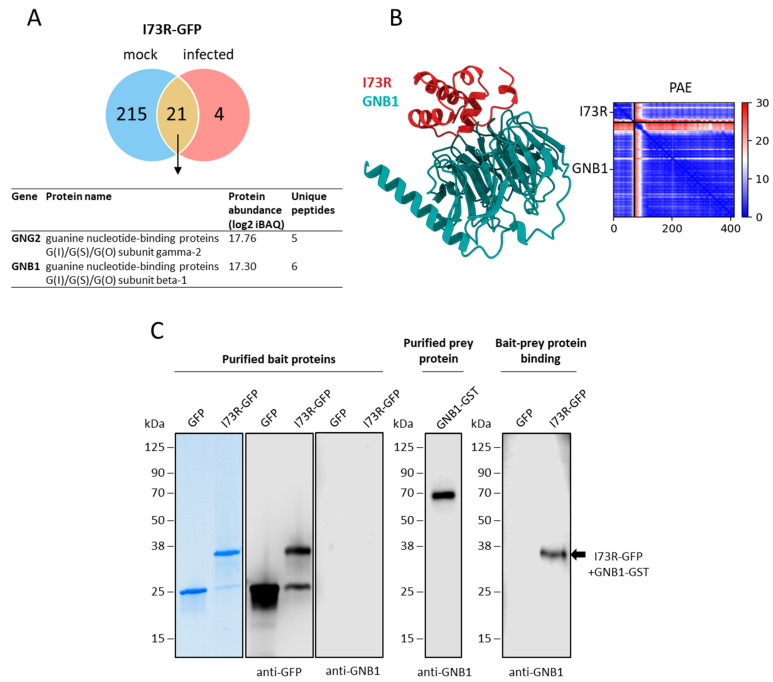
Interaction between pI73R and GNB1. (**A**) The Venn diagram illustrates the partial overlap of the pI73R interactomes observed in non-infected and ASFV-infected cells. The accompanying table highlights two top high-confidence interactors of pI73R, GNB1, and GNG2, which are common to both conditions. (**B**) Predicted structure of the pI73R (red) and GNB1 (green) as ribbon representation. Blue areas with low predicted aligned error (PAE) values indicate potential interactions within and between GNB1 and pI73R. (**C**) Far Western blotting of pI73R-GNB1 complex. GFP-trap purified proteins GFP and GFP-pI73R were separated by SDS-PAGE and transferred onto nitrocellulose membranes. The membrane was incubated with purified GNB1-GST. After washing, bound protein was detected with a GNB1 antibody. The arrow points to a distinct band corresponding to the pI73R-GFP protein indicating specific binding to the GNB1-GST protein.

**Table 1 ijms-26-11768-t001:** Summary of TM-align pairwise structural comparisons using pI73R (PDB: 7VWV) as the query protein.

Reference Protein	Reference PDB ID	Protein Family	TM-Score ^1^	RMSD (Å) ^2^	Aligned Length ^3^
FOXA1	7VOX	FOX family	0.73098	2.16	66
FOXA2	7YZF	FOX family	0.73680	2.30	67
FOXG1	7CBY	FOX family	0.73435	2.36	67
FOXH1	7YZB	FOX family	0.73821	2.74	71
FOXM1	3G73	FOX family	0.72205	2.24	66
FOXK2	2C6Y	FOX family	0.72879	2.01	65
FOXN1	6EL8	FOX family	0.74521	2.39	67
FOXN3	6NCM	FOX family	0.73821	2.04	65
FOXO1	3CO7	FOX family	0.73043	2.50	68
FOXO4	3L2C	FOX family	0.71315	2.45	66
ZBP1	3EYI	Z-DNA binding	0.64462	1.91	58
ADAR1	1QBJ	Z-DNA binding	0.72164	1.31	61
PKZ	4KMF	Z-DNA binding	0.71986	1.28	60
DLM1	1J75	Z-DNA binding	0.65001	1.37	55
ORF112	4HOB	Z-DNA binding	0.62616	2.15	55
E3L	1SFU	Z-DNA binding	0.66341	1.83	62

^1^ TM-score normalized by the length of the query protein. ^2^ Root mean square deviation in Ång ströms. ^3^ Number of pI73R residues matching the reference protein structure.

## Data Availability

All MS raw data and MaxQuant output tables were deposited in the ProteomeXchange Consortium (http://proteomecentral.proteomexchange.org) via the PRIDE partner repository [88] with the dataset identifier PXD066360 and DOI:10.6019/PXD066360 and will be publicly available upon publication.

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
