# Peer review of "Structural and Functional Analysis of ASFV pI73R Reveals GNB1 Binding and Host Gene Modulation"

_ijms, 2025, doi:10.3390/ijms262411768_

Round 1

Reviewer 1 Report

Comments and Suggestions for Authors

This manuscript describes potential functions of the pI73R protein of African swine fever virus (ASFV), a protein shown previously to be dispensable for the virus life cycle (cited, ref 11). First, the authors used various in-silico approaches to describe structural features of pI73R. They confirmed previous data (cited, ref 9) showing a high degree of similarity between pI73R and Z-DNA binding proteins. In addition, they found numerous potential phosphorylation sites and a high degree of similarity with transcription factors of the fork-head box (FOX) protein family for which they provide 3D modelling, suggesting that, like FOX proteins, pI73R may be involved in host gene regulation. They show also that this protein is highly conserved among the four different genotypes I, II, IX and X at least. To further investigate the pI73R protein functions experimentally, they performed expression kinetics and proteomics studies in the wild boar lung-derived macrophage cell line WSL. They confirm previous data showing that this protein is highly expressed in cells early after infection (cited, refs 11 and 36), both for genotype II and genotype IX prototype strains. Differential proteome analysis by mass spectrometry upon pI73R overexpression revealed pI73R-dependent downregulation of CRNKL1, a protein found to bind several proteins of the spliceosome. A pI73R interactome analysis in ASFV-infected versus non-infected host cells expressing pI73R-GFP or GFP alone, along with far Western blot analysis demonstrated a direct interaction between pI73R and the guanine nucleotide-binding proteins subunit beta-1 (GNB1), a protein involved cell signaling and nuclear import. From this the authors conclude that pI73R may rely on GNB1 interaction for nuclear translocation and functions in the regulation of host gene expression.

General evaluation:

This is an interesting study that combined analyses in silico and proteomics experiments in cell culture to suggest potential functions of the ASFV pI73R protein in host gene modulation. The experiments are performed and analyzed correctly. Some conclusions should however be formulated carefully (see below). The data are mostly descriptive, opening avenues for exploration of the different mechanisms suggested. A few simple additional experiments may strengthen the data. With these and the minor issues considered, this study is certainly worth publishing.

Specific comments:

Major issues:

Comment 1: one major drawback of the experiment described in paragraph 2.4. is that the differential proteome analyses are performed out of the ASFV context.  An important question is whether I73R would downregulate CRNKL1 in the context of an ASFV infection too, which is not trivial. A complementary approach to show this would be a differential host proteome analysis with I73R gene-deleted versus parent ASFV. CRNKL1 should be clearly differentially regulated, i.e. downregulated with parent but not with I73R gene-deleted ASFV. If this cannot be done in a reasonable timeframe, this should at least be discussed carefully and conclusions on I73R-mediated downregulation of CRNKL1 should be interpreted with care. It is true that pI73R overexpression downregulates CRNKL1, but it remains to be shown if this occurs and is relevant in the context ASFV infection. This should be discussed clearly.

Comment 2: Experiments looking at I73R localization in infected cells and in cells overexpressing the protein would greatly strengthen the data of this study. Since GNB1 was found to interact with pI73R, which is suggested to facilitate nuclear entry of pI73R (lines 433-438), the authors should look at this in situ. With the plasmid expressing chimeric pI73R-GFP protein versus GFP alone and with the anti-GNB1 antibody, the authors have the possibility to do confocal microscopy for pI73R co-localization with GNB1 and even live cell imaging for pI73R trafficking. The only drawback of the latter approach is that pI73R-GFP may not be transported equally well than pI73R alone.

Minor issues:

Comment 3: on line 32, “lethality” would be more accurate than “mortality”

Comment 4: monocytes (line 39) are typically not susceptible to ASFV. To the best of my knowledge macrophages need to be well differentiated to be susceptible to ASFV. This should be corrected accordingly.

Comment 5: the authors should delete “strain” on line 43 since the genotype II viruses that circulate in Europe, Asia and the Caribbean are not limited to a single strain (according to the standard definition of a virus strain).

Comment 6: The highly conserved amino acid sequence of pI73R among different genotypes with the two main variations Y/H at position 34 and TN/N at the C-terminus were highlighted previously in the supplementary data of reference 11. Therefore, the first sentence of paragraph 2.3. (line 148-149) and lines 364-367 of the discussion should be complemented with something like “as shown previously”, with reference to citation 11.

Comment 7: The authors should revise the statements of lines 351 to 357, especially with respect to the ASFV mRNA. The sentence of lines 351-352 implies that ASFV RNA is produced in the nucleus, which is against the common understanding that ASFV RNA transcription occurs in the cytoplasm, although there is some controversy about this in the literature. But since the authors didn’t look at ASFV mRNA synthesis and trafficking, their speculation of line 352 is not justified (nuclear export of unspliced ASFV RNA). Based on current knowledge, pI73R/CRNKL1 may (if this holds true in the ASFV context, see comment 1) modulate host responses by impairing the splicing/export of host mRNA rather than ASFV mRNA.

Reviewer 2 Report

Comments and Suggestions for Authors

Reviewer comments

The authors in the manuscript entitled “Structural and Functional Analysis of ASFV pI73R Reveals GNB1 Binding and Host Gene Modulation” discusses the structure-functional role of African Swine Fever (ASFV) I73R. Computational prediction revealed DNA binding domains, transcriptional regulatory role in host protein machinery and triggering pathogenesis. The authors have identified host protein Guanine nucleotide-binding protein subunit beta-1 (GNB1) as a novel direct interactor of pI73R which may facilitate its nuclear transport.  The experiments have been well designed and the results presented nicely.

The reviewer has some minor comments to improve the study further if the authors could perform few more additional experiments.

  1. Confocal microscopy to show colocalization between ASFV I37R and host cell protein GNB1.
  2. Knockdown/Knockout experiments to remove the host cell GNB1 expression. What happens to ASFV I37R role in pathogenesis if there is no direct interaction between the viral and the host cell protein.

These two experiments will be an added value to the study.
